# Intrinsic Performance of Monte Carlo Calibration-Free Algorithm for Laser-Induced Breakdown Spectroscopy

**DOI:** 10.3390/s22197149

**Published:** 2022-09-21

**Authors:** Igor B. Gornushkin, Tobias Völker

**Affiliations:** BAM Federal Institute for Materials Research and Testing, Richard-Willstätter-Straße 11, 12489 Berlin, Germany

**Keywords:** laser-induced breakdown spectroscopy, calibration-free analysis, Monte Carlo algorithm

## Abstract

The performance of the Monte Carlo (MC) algorithm for calibration-free LIBS was studied on the example of a simulated spectrum that mimics a metallurgical slag sample. The underlying model is that of a uniform, isothermal, and stationary plasma in local thermodynamical equilibrium. Based on the model, the algorithm generates from hundreds of thousands to several millions of simultaneous configurations of plasma parameters and the corresponding number of spectra. The parameters are temperature, plasma size, and concentrations of species. They are iterated until a cost function, which indicates a difference between synthetic and simulated slag spectra, reaches its minimum. After finding the minimum, the concentrations of species are read from the model and compared to the certified values. The algorithm is parallelized on a graphical processing unit (GPU) to reduce computational time. The minimization of the cost function takes several minutes on the GPU NVIDIA Tesla K40 card and depends on the number of elements to be iterated. The intrinsic accuracy of the MC calibration-free method is found to be around 1% for the eight elements tested. For a real experimental spectrum, however, the efficiency may turn out to be worse due to the idealistic nature of the model, as well as incorrectly chosen experimental conditions. Factors influencing the performance of the method are discussed.

## 1. Introduction

Calibration-free (CF) methods in laser-induced breakdown spectroscopy (LIBS) serve as an alternative to calibration-based techniques [1]. Their major advantage is the ability for rapid chemical analysis without standards; this is important in situations where matrix-matched standards are not available or amounts of samples are limited. However, their typical accuracy and precision are modest and degrade toward semi-quantitative analysis for concentrations below 1%. The main applications of CF LIBS can be found in industry, geology, biology, archeology, or even space exploration [1,2].

Over the years, several approaches to CF LIBS have been proposed, all relying on an assumption of a uniform isothermal plasma at local thermodynamic equilibrium (LTE). They can be divided into two categories: those that use the experimental spectrum to determine the plasma parameters (the inverse problem) and those that specify the plasma parameters to reproduce the experimental spectrum (the direct problem). The most frequently used version of the first category is the Boltzmann plot (BP) or Saha–Boltzmann plot (SBP) method [2,3]. It relies on the assumptions of stoichiometric ablation, plasma uniformity, and optical thinness. The latter condition, though, can be relaxed by the proposed self-absorption correction methods [4]. This method has been repeatedly used to solve specific analytical problems. Praher et al. [5] and Pedarnig et al. [6] paid special attention to the homogeneity and optical transparency of the plasma during a CF LIBS analysis of steel slags and other industrial materials. They carefully optimized the timing of radiation collection and corrected lines for self-absorption. Gornushkin et al. [7] studied effects of non-uniformity of LIBS plasma on the accuracy of calibration-free BP analysis. Hermann et al. [8] took into account the influence of density and temperature gradients on the plasma spectra by assuming two homogeneous zones within the plasma, each with its own density and temperature. Wester and Noll [9] also proposed a heuristic two-shell plasma model to account for plasma gradients.

Several single-point calibration methods have been proposed to improve the CF-BP–LIBS method; they needed only one reference standard with a composition similar to the sample being analyzed. Although not truly calibration-free, they are still close to CF because, in the end, they provide concentrations of elements not included in the certified standard. The methods reduce the error of CF-BP analysis associated with instrumental factors or inaccuracy of spectroscopic data. Cavalcanti et al. [10] demonstrated an improvement in the accuracy of the CF LIBS analysis by first applying it to a certified sample and then to unknown samples. Gaudiuso et al. [11] proposed an inverse CF-LIBS method in which one certified sample was used to find the unique slope of a Boltzmann plot, which was then applied to unknown samples. Aragon and Aguilera [12,13] proposed a one-point calibration “C-sigma” method which was based on the theory of curves-of-growth. Here, several elemental BPs were merged into a single BP that was common for all the elements. Grifoni et al. [14] compared the three one-point calibration methods in favor of the first.

All CF-BP (inverse) methods require solving ill-posed problems of self-absorption correction [15] and deconvolution [16], which can lead to significant errors in the calculation of the integral line intensities (see Equation (1) below). Another disadvantage of these methods is their large error in determining medium and low concentrations at the level of 1% or less. The error is related to the need to sum the element concentrations up to 100% to exclude the experimental factor from the model equations. Because of this, small errors at high concentrations are transformed into large errors at low concentrations. This disadvantage, however, is not inherent in single-point calibration methods that do not use such summation or use it only as part of the procedure [10,11,12,13,14]. Another source of error may be the inadequacy of the model of a homogeneous isothermal plasma to the parameters of a real plasma [17]. Errors can also arise due to uncertainties in spectroscopic parameters and experimental factors, such as neglecting the light-collection geometry [18]. Gornushkin et al. [19] studied the factors of optical density, plasma inhomogeneity, line overlap, noise, spectral resolution, electron density, and path length that affect the accuracy of quantitative analysis by CF-BP–LIBS.

Another group of methods belonging to the second category (direct problem) was proposed by the authors [20,21,22,23]. These methods are based on generating synthetic spectra by using the collision-dominated model of a homogeneous isothermal plasma in LTE (see, for example, Reference [21]) and comparing these spectra with experimental ones. Their advantage is that there are no requirements for optical thinness and deconvolution of overlapping lines; these effects are automatically accounted for when generating synthetic spectra. Yaroshchyk et al. [20] developed an automated standard-free LIBS method that allows for a fast multi-element analysis. The plasma spectrum was represented by a system of simultaneous algebraic equations, which were solved by using the singular value decomposition algorithm. Gornushkin et al. [21] developed a radiation dynamic model of the post breakdown laser plasma in which the problem of determining the concentration of elements was solved by a direct comparison of the calculated synthetic spectra with the experimental ones. Simulated annealing Monte Carlo optimization was used to determine the plasma temperature and density. In a subsequent paper, Herrera et al. [22] compared CF-BP–LIBS with simulated annealing Monte Carlo (MC–LIBS) by analyzing aluminum alloy samples ablated into vacuum. They found that the relative concentrations obtained with CF-BP and MC–LIBS were comparable in magnitude with relative errors of 30–250%. It was argued that the improvement of the spatial and temporal resolution of the experiment is no less important than the refinement of theoretical models. Demidov et al. [23] developed a new MC–LIBS algorithm suitable for graphical processing unit (GPU) computing, in contrast to central processing unit (CPU) computing, which requires unacceptably long processing times. The reduction in computation time was achieved through massive parallel computing on GPUs containing thousands of coprocessors. The performance of MC–LIBS was tested on the spectra of mixtures of metal oxides CaO, Fe_2_O_3_, MgO and TiO_2_, which simulated by-products of metallurgical production, steel slags. Comparison with CF-LIBS showed that the accuracy of the MC–LIBS and CF-LIBS methods is the same for this type of sample.

In the abovementioned MC–LIBS papers [21,22,23], the effectiveness of the method was evaluated by using experimental spectra. Therefore, it was difficult to separate errors due to instrumental factors from errors due to the method itself, for example, due to the stochasticity of the method. In this work, we continued to improve the MC algorithm [23] and study its internal characteristics, which were not considered in previous publications. The reliability of the algorithm was tested on synthetic spectra simulating samples of metallurgical slag. These spectra are free from instrumental factors and completely correspond to the mathematical model of a homogeneous isothermal plasma in LTE. This work should prove that MC–LIBS is a viable alternative to CF-BP–LIBS that can provide higher accuracy of CF analysis for both high and low concentrations.

## 2. Materials and Methods

The plasma is assumed to be isothermal and uniform at local thermodynamic equilibrium (LTE). The radiation along a line of sight is given by the following equation:(1)$$I\left(\lambda ,T\right)=B\left(T\right)\left(1-e^{-\tau \left(\lambda ,T\right)}\right)$$
where I(λ,T) is the spectral intensity, B(T) is the Planck function, and τ(λ,T) is the optical density. The optical density can be factorized into two terms, namely a wavelength-dependent and wavelength-independent optical density:(2)τ(λ,T)=K(T)P(λ)

Here, $K\left(T\right)=\frac{e^{2}\lambda_{0}^{2}}{4\varepsilon_{0}m_{e}c^{2}}f_{ik}\frac{g_{i}e^{-E_{i}/kT}}{U\left(T\right)}\left(1-e^{-\left(E_{k}-E_{i}\right)/kT}\right)nR$ where e and $m_{e}$ are the elementary charge and mass, c and k are the speed of light and Boltzmann constant, $\varepsilon_{0}$ is the permeability of free space, $\lambda_{0}$ is the wavelength in the line center, $g_{i}$ is the degeneracy of the lower transition level, U(T) is the atomic or ionic partition function, $E_{i}$ and $E_{k}$ are the lower and upper transition energies, n is the number density of atomic or ionic species, and R is the radiation path length through the plasma. Term P(λ) represents a line shape function (Gaussian, Lorentzian, or Voigt). In our algorithm, two functions are tested that are typical to LIBS spectra: Lorentzian and Voigt.

A Monte Carlo (MC) algorithm minimizes a cost function that signifies the difference between synthetic and experimental spectra. During the minimization, the physical parameters of the model ($T,R,\;n_{i})$ are varied and gradually approach that of experimental plasma. After finding the minimum, the sought-for parameters of the experimental plasma are read from the model. Mathematically, the problem is expressed by Reference [23]: (3)$$f\left(I^{ex},\;I^{syn}\right)=min_{D}\left[f\left(I^{ex},\;I^{syn}\right)\right]\;\;D=\left\{\begin{array}{c} 10^{13}\leq n_{i}\;\left[{\mathrm{cm}}^{-3}\right]\leq 10^{19},\;\;i=1..N\\ 5000\leq T\;\left[K\right]\leq 20000\\ 0.001\leq R\;\left[\mathrm{cm}\right]\leq 0.1 \end{array}\right\}$$
where $f\left(I^{ex},\;I^{syn}\right)$ is the cost function to be minimized; $I^{ex}$ and $I^{syn}$ are the experimental and synthetic spectra (dependency of $I^{ex}$ and $\;I^{syn}$ upon $\lambda ,n_{i},T$ is omitted); and *D* is the (N+2)-dimensional search domain, where N is the number of chemical elements considered in the model.

A cost function can be any suitable metric that is sensitive to a difference between two sets of data. For example, one can use a correlation coefficient for the construction of the cost function:(4)$$f_{C}\left(I^{ex},\;I^{syn}\right)=1-\frac{\sum \left(I_{i}^{ex}-\bar{I^{ex}\;}\right)\left(I_{i}^{syn}-\bar{I^{syn}}\right)}{{\left[\sum {\left(I_{i}^{ex}-\bar{I^{ex}}\right)}^{2}\right]}^{1/2}{\left[\sum {\left(I_{i}^{syn}-\bar{I^{syn}}\right)}^{2}\right]}^{1/2}}$$

Here, Iiex and Iisyn are the intensities of the experimental and synthetic spectra at wavelength λi, and Iex¯ and Isyn¯ are their corresponding averages over a wavelength range. Function (4) is sensitive to mutual positions and intensities of spectral lines on two compared spectra and depends on the cosine of an angle between two vectors. The vectors represent synthetic and experimental spectra in a multidimensional space, ℝM, where M is the number of spectral points (λi, i=1…M). If the vectors are collinear, the two spectra perfectly match and fC(Iex, Isyn)=0. The contribution of an element to the cost function depends on its concentration and number of spectral lines available for this element. To equalize contributions from all elements, weights (wk) are added in Equation (4):(5)fCW(Iex, Isyn)=1−∑kwk∑LkMk(Iiex−Iex¯)(Iisyn−Isyn¯)[∑kwk∑LkMk(Iiex−Iex¯)2]1/2[∑kwk∑LkMk(Iisyn−Isyn¯)2]1/2

To calculate $w_{k}$, equal weights (W) are first assigned to all elements regardless of their concentrations; W=1/N, with N being the number of elements. The contribution of each spectral line of a given element is then calculated based on the integral intensity of this line divided by the sum of integral intensities of all lines belonging to this element: $w_{np_{n}}=W\cdot (S_{np_{n}}/\underset{p_{n}}{\sum }S_{np_{n}}),\;n=1,\ldots N,\;p_{n}=1\ldots P_{n}$ where 
$S_{np_{n}}$ is the integral intensity of line $p_{n}$ of element n, and $P_{n}$ is the number of lines available for this element. The integral intensities of all lines are recovered from the experimental spectrum. A full spectral grid is split into K spectral fragments so that $L_{k}$ and $M_{k},\;k=1,\ldots K$ denote the lower and upper boundaries of a particular fragment k. Each fragment may contain one or several lines of the same or different elements. The weight of each fragment is then determined by a simple relation: $w_{k}=\underset{n,p_{n}}{\sum }w_{np_{n}}S_{np_{n}}/\underset{n,p_{n}}{\sum }S_{np_{n}}$. The weighting of the cost function equalizes its sensitivity to both weak and strong lines, making it possible to determine trace elements with the similar accuracy as the main ones.

The main concept of the MC–LIBS method is to minimize the cost function, which measures the similarity between model-generated and experimental spectra. This is a non-linear problem (Equation (1)) that needs to be solved in a high-dimensional space (Equation (3)). Standard optimization methods, such as, for example, Newton–Raphson or Levenberg–Marquardt, are inefficient in this case, since the solution can be caught in some local minimum or maximum. Therefore, we use a global optimization algorithm that works as follows.

Many initial random combinations (configurations), $N_{C}$, of plasma parameters ($n_{i}$, T, and R) are taken from hypercube D in Equation (3) and used for generation of $N_{C}$ synthetic spectra. Each configuration is represented by a point in box D that has volume, $V_{D}$, in a phase space of plasma parameters. Values of the cost function are calculated for these initial configurations, and smaller subset $N_{b}$ of points ($N_{b}\ll N_{C}$) is chosen that corresponds to $N_{b}$ smallest values of $f_{CW}\left(I^{ex},\;I^{syn}\right)$. Boxes of smaller volumes, $V_{b}^{\left(1\right)}<V_{D}$, are then built around each such point. In a next iteration, a fraction, $\alpha N_{C}$, of configurations is taken from the original box $V_{D}$, while another fraction, $\left(1-\alpha \right)N_{C}$, is taken from smaller boxes $V_{b}^{\left(1\right)}$, where 0<α<1. After calculating the cost functions for the new set of configurations, new $N_{b}$ points are chosen for the smallest values of the cost function, and new $N_{b}$ boxes are built around those points such that $V_{b}^{\left(2\right)}<V_{b}^{\left(1\right)}<V_{D}$, and so on. As a result, a configuration is found that yields the global minimum of the cost function without a danger of being trapped in local minima. The adjustable parameters of the optimization are the total number of configurations, $N_{C}$; number of boxes, $N_{b}$; rate of reduction of volumes, $V_{b}$; and fraction of configurations, α, taken from small, $V_{b}^{\left(j\right)}$, and large, $V_{D}$, boxes.

To accelerate the described method, a large number of synthetic spectra can be calculated in parallel. The number $N_{c}$ is determined by the available processors. With the help of graphics processing units (GPUs), the parallel calculation of millions of spectra is thus possible.

## 3. Results

The Monte Carlo algorithm is run on a GPU card. A typical GPU works with vectors rather than matrices; matrix operations are prohibited. Therefore, for arrays with dimensionality greater than 1, say L×M×N, only one data dimension, e.g., L, can be simultaneously processed by the GPU (parallelized) while two other dimensions, M and N, are run within a nested loop on the CPU (central processing unit). For efficient operation, the number of such CPU-loops should be maximally reduced. In the proposed MC algorithm, the plasma parameters are calculated on the GPU for ~10^5^–10^6^ simultaneous configurations at each wavelength scanning the whole wavelength range in a loop on the CPU. To reduce time for looping on the CPU, narrow spectral fragments around chosen lines are selected. The selected lines are bracketed within intervals with margins of plus–minus 10–15-line widths from line centers. Such intervals are found to be optimal for reliable subtraction of a baseline. The algorithm is implemented on the @MATLAB 2020b platform, using a desktop 3 GHz PC with a NVIDIA Tesla K40 graphical card.

The baseline subtraction is important for finding true heights and widths of spectral lines that are otherwise affected by a radiation continuum and neighboring lines. This procedure is especially important for weak lines that sit on the wings of strong lines. In this work, a polynomial approximation of the baseline is used [24]. Polynomials of powers from 1 and 10 are sequentially tested to find the optimal one. For each tested polynomial, the baseline is subtracted, and lines are approximated by either a Lorentzian or Voigt function P(λ,Δλ) in the exponent of Equation (6):(6)$$I\left(\lambda ,\Delta \lambda ,K\right)=A\left(1-e^{-KP\left(\lambda ,\Delta \lambda \right)}\right)$$
where A, K, and Δλ are the fitting parameters. The optimal polynomial is determined by the minimal error of the least-mean-square fit.

An observed line profile may consist of several profiles; moreover, an individual atomic or ionic line profile can be broadened by self-absorption and instrumental function. A task is to retrieve original line profiles from the observable spectrum. It is postulated that the width of each line in the iterated synthetic spectrum is equal to the corresponding line width in the observable spectrum, regardless of what the actual parameters of the plasma are. Then the own (not self-absorbed) line width,  Δλ, can be extracted by solving the obvious equation: $2I\left(\lambda_{0}+\Delta \Lambda /2,T\right)=I\left(\lambda_{0},T\right)$, where ΔΛ is the FWHMs (full-width-at-half-maxima) of the experimental line. By combining Equations (1) and (2) with this simple relation, a function is composed:(7)$$F\left(\Delta \lambda \right)=1-2e^{-K\left(T\right)P\left(\lambda_{0}+\Delta \Lambda /2,\Delta \lambda \right)}+e^{-K\left(T\right)P\left(\lambda_{0},\Delta \lambda \right)}$$

The root of this function, F(Δλ)=0, is found numerically by fitting the parameter Δλ (a more detail description can be found in Reference [19]). It is easy to show that the root always exists for the physical interval Δλ∈(0,∞) and P(λ,Δλ) represented by Voigt, Lorentzian, or Gaussian functions. Thus, the found Δλ is the own line width that is free from self-absorption broadening. This approach is used to generate lines with unknown Stark parameters; it is not needed for lines with known Stark parameters.

The inherent performance of the MC method was demonstrated with synthetic spectra. A synthetic spectrum was generated that reproduced a spectrum of the slag sample (Table 1). The major element, calcium, was taken at the concentration 10^16^ cm^−3^; the concentrations of other elements were calculated based on stoichiometric ratios of corresponding oxides.

The plasma was assumed to be uniform and isothermal, with a temperature T = 10,000 K and size R = 0.1 cm. Seventy-four spectral lines were used, which are given in Table 2. All of these lines could be clearly identified on the experimental spectrum from a slag sample. No other criteria for choosing the lines were applied; they could be strong or weak, atomic or ionic, or self-absorbed or optically thin. Spectroscopic data for the lines were taken from the NIST database [25]; the Stark broadening parameters were taken from Griem’s tables [26] and pertinent works from the literature [27,28]. The fragment of the synthetic spectrum is shown in Figure 1 to illustrate its appearance. Complicating effects were added to the spectrum in the form of a finite spectral resolution (84 pix/nm) and random Gaussian noise (0.5% of maximal spectrum intensity). Considering this synthetic spectrum as experimental one, the task was to retrieve the sample composition by the MC method and compare it to the certified values.

The convergence of the MC algorithm to the same value of the cost function is illustrated in Figure 2. The algorithm scanned six orders of magnitude in concentrations, two orders in path length, and 10^4^ K range in temperatures. Starting eleven times from a random configuration, the cost function consistently converged to a minimum 0.0023±0.0001. Running 50 iterations took ~5 min on the GPU, examining 500,000 configurations per iteration.

The certified–found correlation plot for the artificial slag sample is shown in Figure 3, and the corresponding accuracy and precision are given in Table 3. To obtain data given in Figure 3 and Table 3, 11 runs by 50 iterations have been performed for each element, starting each time from a random configuration. Since each run takes ~5 min, the total processing time was 5 min × 11 runs = 55 min. Note that, for a practical problem, such extensive calculations are not required; one run will be enough, because the iterations converge to the same value within ±5%. Convergence can be further improved by refining the algorithm on a denser grid.

The points in Figure 3 show the very small scatter around the 45-degree correlation line, while Table 3 implies that the relative errors for major and minor elements are below 1% (except for silicon). The relative standard deviation is always below 10%, mostly around 1%; it comes from random noise that was imposed on the spectrum and stochastic character of the algorithm. As one sees from Table 3, accuracy and precision are slightly worse for elements represented by only a few lines (e.g., only one line at 390.6 nm was available for Si) or weak lines (e.g., Cr I at 357.9 and 359.3 nm shown in the inset in Figure 1). Note that the accuracy and precision of the MC analysis is the same at both low and high concentrations due to the concentration-independent nature of the method’s errors, unlike the CF-BP method. This favorably distinguishes MC–LIBS from the standard BP method, where accuracy and precision significantly deteriorate toward low concentrations due to the closure condition.

## 4. Discussion

It can be seen that, even for a synthetic spectrum, the result is not 100% accurate. The reasons for this are as follows. During the iterative reconstruction, the broadening parameters (due to the Stark effect) were not known for all lines. These parameters were determined from Equation (5), where the error is dependent on the accuracy of the line width measurement. Thus, as the noise fraction increases, the error also increases. Two other conditions for successful reconstruction are good spectral resolution to distinguish closely spaced lines and a wide spectral range necessary to represent each element with a sufficient number of lines. In principle, only one spectral line per element is required. However, since the spectra are subject to instrumental noise, or the parameters of the spectroscopic lines are not always known with 100% accuracy, it is desirable to include as many suitable lines for each element as possible. The effect of an insufficient number of lines for Si (only one) leads to a larger relative error of its determination (see Table 3).

Experimentally, the above conditions require operation with a wide-range high-resolution spectrometer, such as an echelle. It should also be noted that the spectrometer must be carefully calibrated with a standard light source to account for the uneven spectral response of the light detector.

Further, the Monte Carlo CF method is based on an oversimplified model of laser-induced plasma in LTE, in which plasma is assumed to be (i) isothermal, (ii) uniform, and (iii) stationary, and (iv) light is collected along a single line-of-sight. None of these conditions is fully satisfied in real plasma and experiment. Nevertheless, the assumption can still be a reasonable approximation if the experiment is well-planned. Conditions (i) and (ii) can be met by choosing the correct gating for the detection of LIBS spectra: the start of the gate must be delayed with respect to the laser pulse enough for the plasma to thermalize (i.e., equilibrate translational temperatures of electrons and heavy particles). At the same time, the gate length must be short enough to consider the plasma frozen, but also long enough to collect enough light and not lose sensitivity. Condition (iii), stationarity, is provided, again, by the short gating of plasma emission. Conditions (i)–(iii) assume the use of a gated ICCD camera for light detection rather than a CCD camera. Condition (iv) can be provided by diaphragming the light collection path; a top-view geometry could be the most suitable one.

Thus, despite the very promising results of Monte Carlo analysis with a synthetic spectrum (~1% accuracy and precision), similar results for experimental spectra can turn out to be worse (see, for example, References [22,23]), unless special measures are taken to properly organize the experiment considering the above requirements. It is not difficult to ensure the required experimental conditions, in which case the real efficiency of the method can approach the theoretical one. The relatively long processing time (minutes) of each spectrum can hardly be considered a limitation, given the ever-increasing power of modern computers.

## 5. Conclusions

The performance of the Monte Carlo algorithm for calibration-free LIBS was studied on the example of a synthetic spectrum that mimics a metallurgical slag sample. The algorithm runs in parallel and fully automated on a graphical processing unit (GPU). It can work with any types of lines: self-absorbed or optically thin, affected or not by the instrumental function. The optimization takes several minutes and depends on a type of the GPU and number of elements to be determined. Knowledge of spectroscopic and physical parameters is required for all species involved in calculations. The accuracy and precision are about 1%, the latter due to the stochastic nature of the method and the influence of spectral noise. On experimental spectra, the performance may be worse due to the still idealistic character of the underlying model that requires the uniform, isothermal, and stationary plasma in LTE. The method is, probably, best suited for applications, in which standard-based analysis with reference samples is not feasible and requirements for accuracy and precision are relaxed.

Further development of the MC–LIBS method can be expected in the direction of reducing the computational time by further optimizing the algorithm and incorporating temperature and density gradients into the model, which will introduce only a few new iterable parameters into the algorithm. Work in this direction continues.

## Figures and Tables

**Figure 1 sensors-22-07149-f001:**
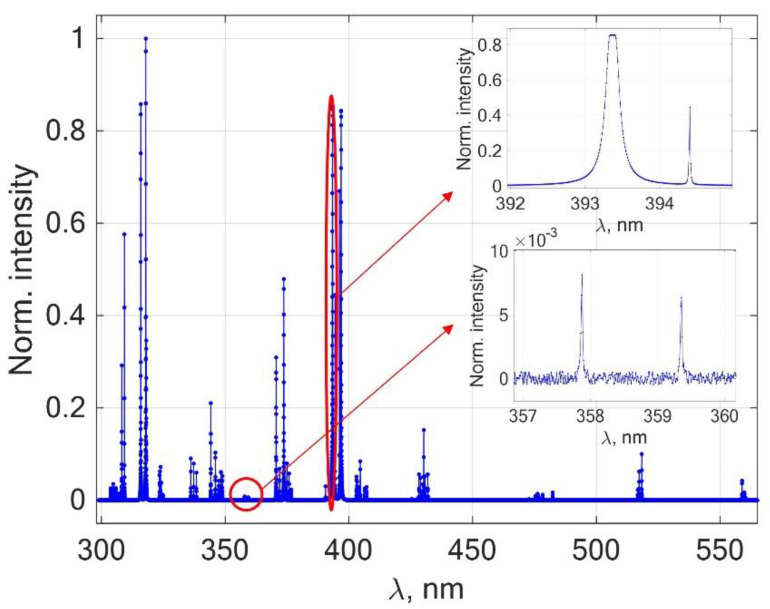
Synthetic spectrum of a metallurgical slag sample.

**Figure 2 sensors-22-07149-f002:**
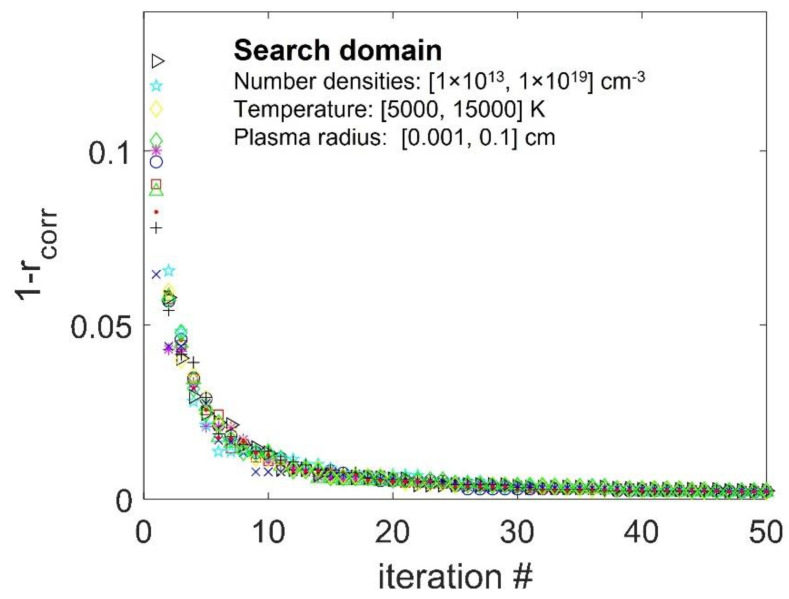
Convergence of the cost function as the function of the number of iterations. Starting from a random configuration, 11 runs were performed. Different symbols correspond to different initial configurations.

**Figure 3 sensors-22-07149-f003:**
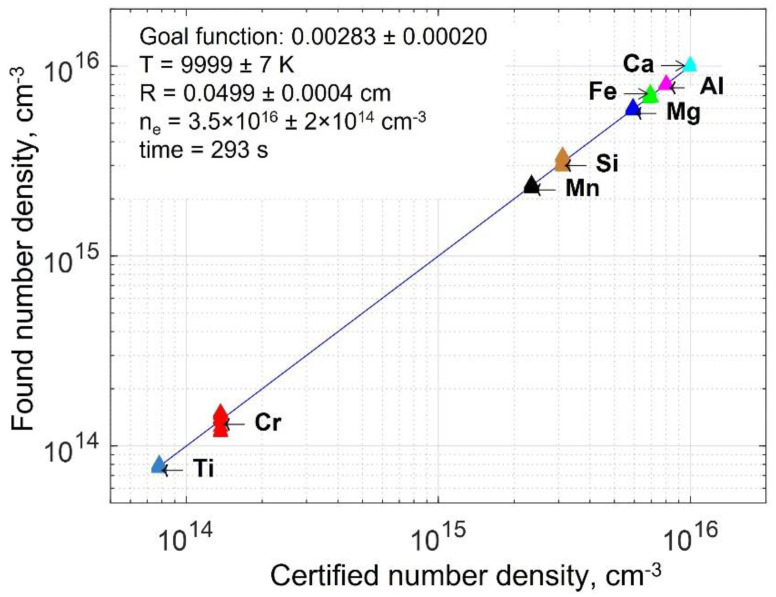
Certified–found correlation plot for synthetic slag spectrum. Each element is represented by 11 colored triangles (some are completely merged) corresponding to 11 runs that started from a random configuration.

**Table 1 sensors-22-07149-t001:** Concentrations of oxides and corresponding number densities used to generate synthetic spectrum.

Oxide	C, % Mass	Element	n, cm^−3^
Al_2_O_3_	19.7	Al	8.03 × 10^15^
CaO	27.0	Ca	1.00 × 10^16^
Cr_2_O_3_	0.5	Cr	1.37 × 10^14^
FeO	24	Fe	6.94 × 10^15^
MgO	11.5	Mg	5.93 × 10^15^
MnO	8	Mn	2.34 × 10^15^
TiO_2_	0.3	Ti	7.80 × 10^13^
SiO_2_	9	Si	3.11 × 10^15^

**Table 2 sensors-22-07149-t002:** Elements and spectral lines used to generate synthetic spectra.

#	Elm	λ nm	E_i_	E_k_	f_ik_	g_i_	g_k_	#	Elm	λ nm	E_i_	E_k_	f_ik_	g_i_	g_k_
1	CaI	428.301	15,210	38,552	0.1990	3	5	38	FeI	404.581	11,976	36,686	0.2120	9	9
2		428.936	15,158	38,465	0.500	1	3	39		406.359	12,561	37,163	0.1650	5	5
3		429.899	15,210	38,465	0.129	3	3	40		407.174	12,969	37,521	0.1900	5	5
4		430.253	15,316	38,552	0.378	5	5	41	MnI	403.075	0	24,802	0.0550	6	8
5		430.774	15,210	38,418	0.1849	3	1	42		403.307	0	24,788	0.0403	6	6
6		431.865	15,316	38,465	0.1200	5	3	43		403.449	0	24,779	0.0257	6	4
7		558.876	20,371	38,259	0.2317	7	7	44		403.575	17,282	42,054	0.0800	8	6
8		559.849	20,335	38,192	0.2009	3	3	45		404.136	17,052	41,789	0.193	10	10
9		560.285	20,349	38,192	0.0399	5	3	46		472.746	23,549	44,696	0.057	6	6
10	CaII	315.887	25,192	56,839	0.847	2	4	47		473.909	23,720	44,815	0.081	4	4
11		317.933	25,414	56,858	0.8200	4	6	48		475.404	18,402	39,431	0.137	6	8
12		318.128	25,414	56,839	0.088	4	4	49		475.585	23,720	44,696	0.210	4	6
13		370.602	25,192	52,167	0.172	2	2	50		476.151	23,819	44,815	0.364	2	4
14		373.690	25,414	52,167	0.164	4	2	51		476.237	23,297	44,289	0.333	8	10
15		393.366	0	25,414	0.682	2	4	52		476.642	23,549	44,523	0.210	6	8
16		396.847	0	25,192	0.330	2	2	53		478.343	18,531	39,431	0.138	8	8
17	AlI	308.215	0	32,435	0.1670	2	4	54		482.352	18,705	39,431	0.139	10	8
18		309.271	112	32,437	0.1570	4	6	55	MnII	344.199	14,326	43,371	0.0500	9	7
19		394.401	0	25,348	0.1160	2	2	56		346.031	14,594	43,485	0.0334	7	5
20		396.152	112	25,348	0.1160	4	2	57		347.413	14,781	43,557	0.0179	5	3
21	MgI	516.732	21850	41,197	0.1350	1	3	58		348.290	14,781	43,485	0.0291	5	5
22		517.268	21870	41,197	0.1350	3	3	59		348.868	14,901	43,557	0.0385	3	3
23		518.360	21911	41,197	0.1360	5	3	60	TiI	498.173	6843	26,911	0.2900	11	13
24	SiI	390.552	15394	40,992	0.091	1	3	61		499.107	6743	26,773	0.2670	9	11
25	FeI	303.739	888	33,802	0.0671	3	5	62		499.950	6661	26,657	0.254	7	9
26		304.760	704	33,507	0.0533	5	7	63	TiII	323.452	393	31,301	0.2685	10	10
27		305.745	6928	39,626	0.0359	11	9	64		323.657	226	31,114	0.2150	8	8
28		305.909	416	33,096	0.0294	7	9	65		323.904	94	30,959	0.198	10	10
29		306.724	7376	39,970	0.0342	9	7	66		324.199	0	30,836	0.2317	4	4
30		371.993	0	26,875	0.0411	9	11	67		324.860	10025	40,798	0.388	6	8
31		372.256	704	27,560	0.0103	5	5	68		336.121	226	29,968	0.3350	8	10
32		374.336	7986	34,692	0.0328	5	3	69		337.279	94	29,735	0.3210	6	8
33		374.556	704	27,395	0.0339	5	7	70		338.376	0	29,544	0.3580	4	6
34		374.826	888	27,560	0.0321	3	5	71	CrI	357.869	0	27,935	0.3660	7	9
35		374.948	7377	34,040	0.1610	9	9	72		359.349	0	27,820	0.2910	7	7
36		375.823	7728	34,329	0.1340	7	7	73		425.434	0	23,499	0.1100	7	8
37		376.719	8155	34,692	0.1360	3	3	74		427.480	0	23,386	0.0842	7	7

**Table 3 sensors-22-07149-t003:** Accuracy and precision of MC analysis; % concentrations of elements are recalculated from concentrations of oxides given in Table 1.

	Ca	Al	Mg	Si	Fe	Mn	Ti	Cr
concentration in wt.%	27.32	22.01	16.16	8.52	19.03	6.35	0.22	0.4
rel. error in %	0.21	0.07	0.94	3.74	0.28	0.04	0.16	0.73
RSD in %	0.73	1.28	2.15	4.0	1.56	1.79	1.72	7.43

## Data Availability

Data are available from authors upon request.

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
