# Peer review of "Intrinsic Performance of Monte Carlo Calibration-Free Algorithm for Laser-Induced Breakdown Spectroscopy"

_sensors, 2022, doi:10.3390/s22197149_

Round 1

Reviewer 1 Report

The reviewer thinks the topic of this manuscript is suitable to be published in Sensors. The topic of this paper is a Monte Carlo CF method and studying its intrinsic characteristics on the example of a simulated spectrum. This method provides better accuracy and precision for calibration-free analysis. However, before publication, there are also some suggestions to the authors.

1. Abstract

1) Page 1 Line 21.

LIBS is an abbreviation for laser-induced breakdown spectroscopy, it is recommended that LIBS not be a separate keyword.

2. Introduction

1) Page 2-3.

The reviewer does not suggest using the serial number of the reference as a noun followed by the preposition.

3. Materials and Methods

The format of this part of the formula does not seem to be in line with the specification and it is essential to consult the template document again.

4. Results

1) Figure 1.

  It would be clear if the author could first circle the large image when zooming in partially, and then use arrows to introduce it to the small image.

2) Figure 2.

  Due to the header being a bit blurry, the reviewer thinks the quality of the image needs to be improved.

3) Page 7 Line 229

  The author mentioned that “11 runs were done starting from a random configuration.” The cost function also converges when the number of iterations is 11. However, it lacks a corresponding description for this result.

4) Figure 3.

  First, whether the latter elements are too close to each other for easy identification, it is suggested that they can be staggered left and right. Second, the shapes of the graphs representing the Cr and Si elements are not consistent with the others, and the relevant descriptions are missing from the graph names. Third, the units of the horizontal and vertical coordinates of the image are missing and the objective function is not written in a standard way.

5. Discussion

1) Page 8 Line 278.

Table 3 does not reflect the processing time, and the reader may not understand the exact time required. It is suggested that this data can be reflected and then explained accordingly.

Author Response

Answers to reviewers

We thank the reviewers for their time and valuable comments that help us to improve the manuscript. Changes and additions to the text are given under the reviewer's comments (reproduced in italic). New fragments in the text are marked in red.

Reviewer 1

The reviewer thinks the topic of this manuscript is suitable to be published in Sensors. The topic of this paper is a Monte Carlo CF method and studying its intrinsic characteristics on the example of a simulated spectrum. This method provides better accuracy and precision for calibration-free analysis. However, before publication, there are also some suggestions to the authors.

We do our best to resolve issues raised by the reviewer.

  1. Abstract

1) Page 1 Line 21.

LIBS is an abbreviation for laser-induced breakdown spectroscopy, it is recommended that LIBS not be a separate keyword.

LIBS is excluded from the keywords.

  1. Introduction

1) Page 2-3.

The reviewer does not suggest using the serial number of the reference as a noun followed by the preposition.

We've revised the quote format; Author names now precede citations.

  1. Materials and Methods

The format of this part of the formula does not seem to be in line with the specification and it is essential to consult the template document again.

We consulted the template provided at https://www.mdpi.com/journal/sensors/instructions  and found that our description of the method generally agreed with the instructions. The respected reviewer mentions the formula but does not specify which one. We are puzzled by this comment and therefore are not taking any action on it.

  1. Results

1) Figure 1.

  It would be clear if the author could first circle the large image when zooming in partially, and then use arrows to introduce it to the small image.

Done, thank you.

2) Figure 2.

  Due to the header being a bit blurry, the reviewer thinks the quality of the image needs to be improved.

We have improved Fig.2

3) Page 7 Line 229

  The author mentioned that “11 runs were done starting from a random configuration.” The cost function also converges when the number of iterations is 11. However, it lacks a corresponding description for this result.

It is not right. The number of runs is 11 and the number of iterations in each run is 50. This can be easily deduced from both Fig. 2 (where the X-axis scale ends at #50) and from the text “50 iterations took ~ 5 minutes…”. We are not taking any action on this comment.

4) Figure 3.

  First, whether the latter elements are too close to each other for easy identification, it is suggested that they can be staggered left and right. Second, the shapes of the graphs representing the Cr and Si elements are not consistent with the others, and the relevant descriptions are missing from the graph names. Third, the units of the horizontal and vertical coordinates of the image are missing and the objective function is not written in a standard way.

We have taken care of all three issues raised by the reviewer regarding Fig.3. We have explained the shapes of the symbols in the figure caption.

  1. Discussion

1) Page 8 Line 278.

Table 3 does not reflect the processing time, and the reader may not understand the exact time required. It is suggested that this data can be reflected and then explained accordingly.

Thank you for this comment. We included the total processing time required to obtain data in Table 3 and Fig.3. A new fragment is added below Fig.2 starting with “ Since each run takes ~5 minutes, the total processing time was…”

Reviewer 2 Report

In this work the authors use the Monte Carlo (MC) approach using a graphic processing unit (GPU) to analyze the emission of a synthetic plasma produced on metallurgical slag samples. The same group already presented  this same methodology to investigate real samples (ref 11: Spectrochim Acta , 125 (2016) 97-102 ). In that work, the authors showed that the Monte Carlo standardless approach, using a GPU, obtains similar results to those obtained with calibration free LIBS.  In that work, where real spectra are used, the errors are much larger than those presented in this manuscript.

The modeling presented here shows some interesting results, so I consider that this paper is interesting for the LIBS community.  However, the authors should show more clearly what is the contribution of this work with respect to what has already been published. This should be discussed in the introduction of the article.

In the discussion section it is mentioned that the simulation assumes several simplifications (line 261), several of them can be achieved using an ICCD camera and an echelle spectrometer. However, one of the main advantages of the LIBS technique is its portability and an ICCD + echelle is not a handheld device. Normally on-site analysis requires the use of a miniature CCD spectrograph. The authors should further discuss the potential of the proposed method by using a portable detector where the emission is integrated in time. I recommend further discussion of the applicability of this method to more complex samples such as soils and the ability to predict minority and trace elements as compared to the normal Calibration Free LIBS method.

Author Response

Reviewer 2

In this work the authors use the Monte Carlo (MC) approach using a graphic processing unit (GPU) to analyze the emission of a synthetic plasma produced on metallurgical slag samples. The same group already presented this same methodology to investigate real samples (ref 11: Spectrochim Acta , 125 (2016) 97-102 ). In that work, the authors showed that the Monte Carlo standardless approach, using a GPU, obtains similar results to those obtained with calibration free LIBS.  In that work, where real spectra are used, the errors are much larger than those presented in this manuscript.

The modeling presented here shows some interesting results, so I consider that this paper is interesting for the LIBS community.  However, the authors should show more clearly what is the contribution of this work with respect to what has already been published. This should be discussed in the introduction of the article.

The actual spectra that we used in our previous publication (old ref 11, new 22) do not fully satisfy the assumptions of the models, since a real plasma is never homogeneous or isothermal. In addition, there are always errors due to spectrometer calibration and instrumental noise. In this article, we deliberately use synthetic spectra, which are fully consistent with the mathematical model, to study intrinsic characteristics of the method itself. We also use an improved cost function minimization algorithm and describe it in detail. We have reshuffled the Introduction and added a new large fragment (marked in red) that reviews earlier MC LIBS publications and highlights the difference between this work and previous ones.

In the discussion section it is mentioned that the simulation assumes several simplifications (line 261), several of them can be achieved using an ICCD camera and an echelle spectrometer. However, one of the main advantages of the LIBS technique is its portability and an ICCD + echelle is not a handheld device. Normally on-site analysis requires the use of a miniature CCD spectrograph. The authors should further discuss the potential of the proposed method by using a portable detector where the emission is integrated in time. I recommend further discussion of the applicability of this method to more complex samples such as soils and the ability to predict minority and trace elements as compared to the normal Calibration Free LIBS method.

We did not set ourselves the goal of miniaturizing the spectrometer to hand-held size. All calibration-free methods will be unreliable when using portable instruments that are not capable of providing high spectral resolution and time gating. These characteristics are important for satisfying the assumptions of the model (homogeneity, stationarity) that the experimental spectrum must satisfy. We do not change the text in relation to this comment, as it is not relevant to this work.

In this work, we also do not deal with real spectra, but only with synthetic ones. Although we mention that the spectra mimic slag, it could equally well be soil or any other material; the results of this work are applicable to any matrix. But the reviewer is right, analysis of trace elements (say, below 0.1%) can be problematic due to the lack of sensitivity of the cost function to weak lines. But weighing the cost function by eq. 5 should help. We add a small remark on this issue “The weighting of the cost function …” in section 2 below eq. 5.

Reviewer 3 Report

Monte Carlo LIBS (MC-LIBS) is a potential calibration-free LIBS technique. In this manuscript, the authors used synthetic LIBS spectra to test the performance of MC-LIBS and the results showed that the intrinsic accuracy of the MC-LIBS is found to be around 1%. The issue when MC-LIBS was used for a real experimental spectrum was discussed in detail, which is significant for the further development of MC-LIBS. The manuscript was well organized and the methods were described completely. My opinion is that this manuscript can be considered for publication in Sensors after minor revision. Detailed comments are as below:

1.      Line 34, I think that “The most frequently used is a version of CF LIBS based on the Boltzmann plot (BP) or Saha-Boltzmann plot (SBP) method” is better. (https://doi.org/10.1016/j.trac.2022.116618)

2.      In Figure 2, the lines with various kinds of symbols need labels.

3.      Table 2 has insufficient resolution, please improve.

4.      In Conclusions, what MC-LIBS needs to be further developed could be described briefly.

Author Response

Reviewer 3

Monte Carlo LIBS (MC-LIBS) is a potential calibration-free LIBS technique. In this manuscript, the authors used synthetic LIBS spectra to test the performance of MC-LIBS and the results showed that the intrinsic accuracy of the MC-LIBS is found to be around 1%. The issue when MC-LIBS was used for a real experimental spectrum was discussed in detail, which is significant for the further development of MC-LIBS. The manuscript was well organized and the methods were described completely. My opinion is that this manuscript can be considered for publication in Sensors after minor revision. Detailed comments are as below:

1.      Line 34, I think that “The most frequently used is a version of CF LIBS based on the Boltzmann plot (BP) or Saha-Boltzmann plot (SBP) method” is better. (https://doi.org/10.1016/j.trac.2022.116618)

Thank you. We modified this line. Also thank you for the reference, we included it in the citation list.

2.      In Figure 2, the lines with various kinds of symbols need labels.

We specified the meaning of the symbols in the figure caption.

3.      Table 2 has insufficient resolution, please improve.

We improved the resolution of this table.

4.      In Conclusions, what MC-LIBS needs to be further developed could be described briefly.

We have added a relevant note to the Conclusions, starting with “Further development of the MC LIBS method can be expected in the direction of…”

Reviewer 4 Report

The manuscript titled "Intrinsic Performance of Monte Carlo Calibration-Free Algorithm for Laser Induced Breakdown Spectroscopy" by Gornushkin and Volker reports a new calibration-free algorithm for quantitative analysis using laser-induced breakdown spectroscopy (LIBS). The authors provided a quite complete review on the currently used CF-LIBS methods. CF-LIBS does not require matrix-matched standards, so it is useful for assessing elemental compositions of complex industrial materials such as slags, ores, etc. However, for the successful (or acceptable) applications of CF-LIBS, local thermodynamic equilibrium and optical thinness need to be acquired. They are known to be the main factors causing deterioration of accuracy. Actually, the Monte Carlo CF-LIBS algorithm proposed by the authors is a spectral reconstruction method by minimizing difference from a reference spectrum. One of the merits of the authors method seems that the self-absorption process is implemented in the spectral intensity model. Thus, the optical thinness condition is released. This is a strong point that should be emphasized. In my opinion, this manuscript is worthy of publication in the journal of Sensors. However, it can be improved considering the following suggestions.

1. In the Introduction section, the authors can provide an easy introduction of the Monte Carlo algorithm briefly to help experimentalists understand the theory. 

2. How many lines are necessary for each elements? To reconstruct spectra, this model also has to calculate excitation and ionization temperatures and electron density. I am curious about the minimum number of lines that can provide reliable results.

Author Response

Reviewer 4

The manuscript titled "Intrinsic Performance of Monte Carlo Calibration-Free Algorithm for Laser Induced Breakdown Spectroscopy" by Gornushkin and Volker reports a new calibration-free algorithm for quantitative analysis using laser-induced breakdown spectroscopy (LIBS). The authors provided a quite complete review on the currently used CF-LIBS methods. CF-LIBS does not require matrix-matched standards, so it is useful for assessing elemental compositions of complex industrial materials such as slags, ores, etc. However, for the successful (or acceptable) applications of CF-LIBS, local thermodynamic equilibrium and optical thinness need to be acquired. They are known to be the main factors causing deterioration of accuracy. Actually, the Monte Carlo CF-LIBS algorithm proposed by the authors is a spectral reconstruction method by minimizing difference from a reference spectrum. One of the merits of the authors method seems that the self-absorption process is implemented in the spectral intensity model. Thus, the optical thinness condition is released. This is a strong point that should be emphasized. In my opinion, this manuscript is worthy of publication in the journal of Sensors. However, it can be improved considering the following suggestions.

We thank the reviewer for the positive assessment of our work. We emphasized the weak point of CF-LIBS with the phrase starting with the words “All CF-BP (inverse) methods require solving ill-posed problems of self-absorption correction [15] and deconvolution [16]…” (with two new citations) and the strength of the MC method with the phrase, starting with "Their advantage is that there are no requirements for optical thinness and deconvolution of overlapping lines ..."

  1. In the Introduction section, the authors can provide an easy introduction of the Monte Carlo algorithm briefly to help experimentalists understand the theory. 

We have significantly expanded the introduction by giving a more detailed description of the Monte Carlo method. The new fragment begins with the words “Another group of methods belonging to the second category (direct problem) has been proposed…”

  1. How many lines are necessary for each elements? To reconstruct spectra, this model also has to calculate excitation and ionization temperatures and electron density. I am curious about the minimum number of lines that can provide reliable results.

In principle, only one spectral line per element is required. However, since the spectra are subject to instrumental noise or the parameters of the spectroscopic lines are not always known with 100% accuracy, it is recommended to include as many suitable lines as possible for each element. Therefore, this question cannot be answered in a general way. A corresponding sentence has been included in first paragraph of Discussion.
